# Mono a Mano: ZBP1’s Love–Hate Relationship with the Kissing Virus

**DOI:** 10.3390/ijms23063079

**Published:** 2022-03-12

**Authors:** Alan Herbert, Aleksandr Fedorov, Maria Poptsova

**Affiliations:** 1InsideOutBio, 42 8th Street, Charlestown, MA 02129, USA; 2Laboratory of Bioinformatics, Faculty of Computer Science, National Research University Higher School of Economics, 11 Pokrovsky Bulvar, 101000 Moscow, Russia; anfedorov@edu.hse.ru (A.F.); mpoptsova@hse.ru (M.P.)

**Keywords:** Z-RNA, Z-DNA, ADAR1, ZBP1, flipons, Epstein–Barr virus, autoimmune disease, systemic lupus erythematous, cancer, lymphoma, exhausted T cells

## Abstract

Z-DNA binding protein (ZBP1) very much represents the nuclear option. By initiating inflammatory cell death (ICD), ZBP1 activates host defenses to destroy infectious threats. ZBP1 is also able to induce noninflammatory regulated cell death via apoptosis (RCD). ZBP1 senses the presence of left-handed Z-DNA and Z-RNA (ZNA), including that formed by expression of endogenous retroelements. Viruses such as the Epstein–Barr “kissing virus” inhibit ICD, RCD and other cell death signaling pathways to produce persistent infection. EBV undergoes lytic replication in plasma cells, which maintain detectable levels of basal ZBP1 expression, leading us to suggest a new role for ZBP1 in maintaining EBV latency, one of benefit for both host and virus. We provide an overview of the pathways that are involved in establishing latent infection, including those regulated by MYC and NF-κB. We describe and provide a synthesis of the evidence supporting a role for ZNA in these pathways, highlighting the positive and negative selection of ZNA forming sequences in the EBV genome that underscores the coadaptation of host and virus. Instead of a fight to the death, a state of détente now exists where persistent infection by the virus is tolerated by the host, while disease outcomes such as death, autoimmunity and cancer are minimized. Based on these new insights, we propose actionable therapeutic approaches to unhost EBV.

## 1. Introduction

Biological roles for the left-handed Z-form nucleic conformation (ZNA, where ZNA represents Z-RNA, Z-DNA, Z-XNA or other left-handed helices with modified bases or spines) have been recently confirmed. ZNA forms from right-handed DNA by flipping the bases over, producing the zig-zag backbone characteristic of the structure (Figure 1A). ZNA is a higher energy conformation most easily formed by sequences of alternating purines and pyrimidines with d(GC)_n_, d(CA)_n_ and d(GT)_n_ being the most favorable (reviewed in [1,2]). ZNA is recognized by the structure-specific family of Zα domain proteins that include the p150 isoform of the double-stranded RNA (dsRNA) editing enzyme ADAR1 (encoded by ADAR) and Z-DNA binding protein 1 (ZBP1) (Figure 1B) [3]. ADAR1 p150 negatively regulates type I interferon responses in both human and mice, reducing the risk of autoimmune diseases such as Aicardi Goutières Syndrome (AGS) [4,5]. It also competes for Z-RNA to prevent activation of ZBP1 dependent pathways (Figure 1B).

ZBP1 can initiate cell death through the regulated cell death (RCD) [6] pathway of apoptosis that is a normal part of development and removes cells without a trace. ZBP1 can also induce the inflammatory cell death (ICD) pathway of necroptosis to stimulate adaptive immune responses to pathogens and cancers [6,7,8,9,10,11,12,13,14] (Figure 1B). Both forms of cell death depend on interactions between RIPK1 (receptor interacting protein kinase 1) and RIPK3, but only RIPK3 kinase activity is required for ZBP1 induced ICD [15]. In both RCD and ICD, the outcomes depend on the recognition of ZNA by the Zα domain, regardless of the nucleotide sequence of the ZNA forming element [16,17]. ZBP1 is also expressed in normal human tissues, suggesting that this protein may regulate other cellular pathways (Figure 1C).

Here, we adopt a systems biology approach. We describe different roles for ZBP1 in calibrating the responses of proliferating T cell to endogenous RNAs and in the neutralization of emerging threats by tissue resident T cells. We also propose a novel role for ZBP1 in chronic viral infection where Z-DNA facilitates switching of epigenetic state to maintain viral latency. Our focus is on Epstein–Barr virus (EBV), a pathogen that causes significant human morbidity and mortality including autoimmune diseases such as systemic lupus erythematosus, multiple sclerosis and cancer [18,19,20,21]. We explore therapeutic approaches useful in preventing or resolving persistent EBV infection.

**Figure 1 ijms-23-03079-f001:**
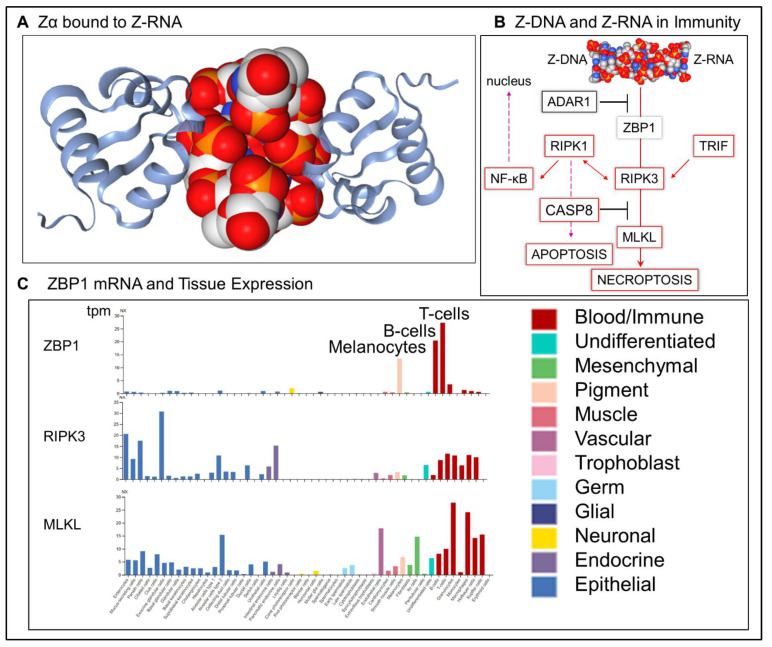
Z-DNA binding protein 1 (ZBP1) has different roles in development of the immune system. (**A**). Structure of the Z-DNA binding protein Zα2 domain bound to the zig-zag backbone of left-handed Z-RNA (from PDB:3EY1 rendered by NGL Viewer [22]). (**B**) ZBP1 senses left-handed Z-DNA and Z-RNA to initiate cell death, either by apoptosis or necroptosis. Both pathways depend on an interaction of receptor interaction protein kinase I and RIPK3. RIPK3 can also be activated by TRIF (toll-like receptor adaptor molecule 1 encoded by TICAM1). Execution of apoptosis depends on caspase 8 activation (CASP8), a protein that also inhibits RIPK3 activation of MLKL( mixed lineage kinase domain-like pseudokinase). ADAR1 (adenosine deaminase RNA specific) inhibits activation of ZBP1 through its Zα domain). RIPK1 is also able to activate the nuclear factor kappa B (NF-κB) that then translocates to the nucleus. (**C**) Expression of ZBP1 in normal tissues is highest in T cells, B cells and melanocytes as measured by tpm (transcripts per kilobase million).

## 2. ZBP1 and the Single Cell

In normal tissue, ZBP1 is expressed at detectable basal levels mostly in T and B cells, with some expression in melanocytes, suggesting a homeostatic role for ZBP1 in the absence of infection and inflammation (Figure 1C). To further explore this possibility, we analyzed gene expression in a published single-cell RNA (scRNA) dataset derived from blood, liver and spleen samples collected from three human samples [23] (Figure 2). We reclustered the data to produce maps distinguishing tissue of origin and cell lineage (Figure 2A,B). Two classes of ZBP1 expressing T cells exist with those expressing the proliferation marker MKI67 distinct from those expressing CD69 mRNAs characteristic of tissue resident cells (TRCs) [24].

To further understand the difference between these TRCs and dividing T-cell types, we analyzed coexpression of gene pairs (Figure 2D), using a color scale for expression of one gene and a size scale to represent expression of the other. The smallest colored dots indicate no coexpression. By plotting both color and size measures for a gene of interest, as shown in the right hand panel for each mRNA analyzed, we could measure the maximum coexpression possible for the gene paired with any other gene. This approach allowed us to identify the cell types expressing ZBP1. These include T cells expressing T-cell receptor (*tcr*) mRNAs encoding the α (TRAC) and δ (TRDC) chain constant regions and NK cells expressing fibroblast growth factor binding protein 2 (FGFBP2) mRNA. The analysis also revealed ZBP1 is coexpressed in one class of T cells with the proliferation marker MKI67 (Figure 2D), while in another class of T cells it is coexpressed with the tissue resident markers CD69 and CXCR6 [24]. In contrast, ZBP1 is not coexpressed with RIPK3 and MLKL mRNAs, indicating that ZBP1-dependent cell death pathways are not constitutively active in normal cells (Figure 1B). ZBP1 instead may be performing other roles.

**Figure 2 ijms-23-03079-f002:**
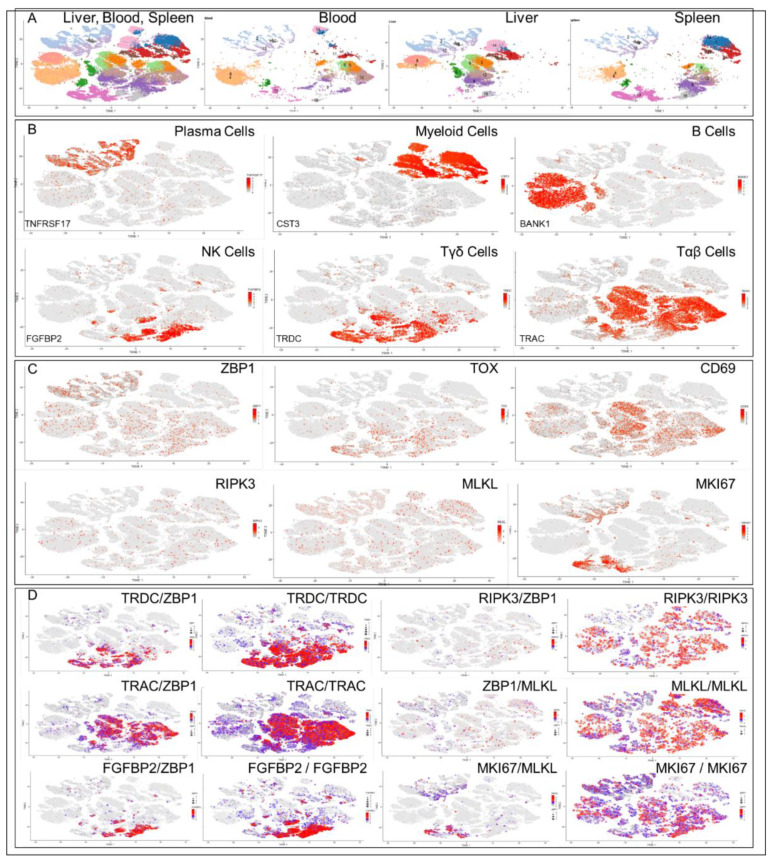
Single cell analysis of RNA expression in human liver, spleen and blood [23] using the UMAP algorithm implemented in the R-package *scater* [25] (random seed = 1000). (**A**). Clusters by cell type. (**B**) Markers used to assign cell type are listed in the lower left-hand corner (**C**). Expression of genes relevant to ZBP1 induced necroptosis (ZBP1, RIPK3, MLKL), development (TOX, thymocyte selection associated high mobility group box), proliferation (MKI67, marker of proliferation Ki-67) and tissue residence (CD69). (**D**). Coexpression of RNAs. The expression level of one gene is indicated by color and that of the other by size. The left panel of each pair is based on two different genes allowing visualization of how often the two genes are coexpressed. The right panel is based on both the size and color of a single gene, giving the maximum coexpression that is possible. (TRAC, T-cell receptor alpha constant region; TRDC, T-cell receptor delta constant region; FGFBP2, fibroblast growth factor binding protein 2). The dataset for this analysis is available in SingleCellExperiment format https://rdrr.io/github/LTLA/scRNAseq/man/ZhaoImmuneLiverData.html (accessed on 7 September 2021) and analyzed using the protocols detailed at https://bioconductor.org/packages/release/bioc/vignettes/scater/inst/doc/overview.html (accessed on 7 September 2021).

## 3. A Model for the Effects of ZBP1 on T-Cell Immunity

A model incorporating the current findings is presented in Figure 3, where the T-cell responses regulated by ZBP1 vary with the stage of T-cell development and differ between those younger, proliferating T cells and the more mature TRC subset. In developing and proliferating cells, we propose that ZBP1 sets a threshold to prevent activation of T cells by ZNA forming self-RNAs. Cells with a high amount of ZNA, either due to endogenous retroelements, aberrant gene transcription or during *tcr* rearrangement, are eliminated by ZBP1-induced, caspase 8 (encoded by CASP8) dependent apoptosis. This process calibrates the immune system response to threats and canalizes expression of parental alleles in the surviving cells [26]. In newly formed T cells, the process also removes cells in which ADAR1 p150 expression is insufficient to prevent *tcr* dependent activation of ZBP1 (Figure 1B).

In TRCs, ZBP1 triggers ICD rather than RCD. ICD is initiated by TRC receptors that are specific for PAMPs (pathogen associated molecular recognition patterns), DAMPs (damage associated molecular recognition patterns) and LAMPs (life-style associated molecular recognition patterns) [27] rather than antigen-specific *tcr*. These TRCs enable an early warning system for detecting emerging threats. They trigger the alarm by undergoing ICD. Their self-sacrifice clears the battlefield for a wave of newly activated, less mature lymphocytes to mount their attack in an antigen-specific fashion. The interferon produced by these cells amplifies the response by inducing ZBP1 expression in other resident TRCs, promoting additional rounds of ICD. Interferon also increases ZBP1 expression in the newly recruited cells and helps shape the immune response. Initially, ICD of young cells enhances the attack, while at later stages, RCD would favor resolution of the response [28,29]. Differentiation of the responding cells leads to memory cell formation and replenishes the stock of TRC effectors [30].

## 4. Are “Exhausted” T Cells Actually Highly Functional TRCs?

TRCs are non-proliferative lymphocytes and are activated by pattern recognition rather than by specific antigens. They also express TOX mRNA. They therefore meet the criteria to classify them as “tired” or “exhausted” T cells (Figure 2, panel C) [30,31]. However, the “tired” TRCs we describe here are not dysfunctional. Rather, these cells are active elements of the immune system tasked with detecting and responding to emerging threats.

Collectively, the ZBP1 dependent cell death pathways enable a number of different schemes for optimizing innate immune responses to protect the host. Both apoptosis and necroptosis remove potentially dangerous cells that threaten survival, with the ICD nuclear option deployed only as a last resort. The different roles played by each cell death pathway in the immune response is most apparent in the early stages of viral and bacterial infection, where a balance between RCD and ICD is necessary to properly calibrate the response [32]. ICD initiated by tissue resident T cells is part of a surveillance system that provides an early warning of emerging threats.

## 5. Mouse Genetic Studies Support the T-Cell Model

Mouse genetic studies offer support for the model detailed in Figure 3, evidencing different roles for ZBP1 dependent RCD and ICD. RCD is important in thymocyte development. Mice with knockout of the ADAR p150 gene in CD4^+^ thymocytes have impaired positive selection of T cells. This outcome likely reflects a high rate of ZBP1-induced RCD due to the accumulation of Z-RNA following T-cell rearrangement and *tcr* driven clonal expansion. The ADAR p150 deficiency persists when the surviving CD4^+^ cells migrate to the periphery. There, during an inflammatory response, higher levels of ZBP1 than in the thymus will be induced by the interferon produced by activated immune cells. Due to the gene knockout in CD4^+^, there will be no compensatory increased production of ADAR1 p150 to downregulate the ICD pathway as exists in wildtype mice. Consequently, the risk of autoimmune disease is increased. Indeed, CD4-p150 deficient mice develop inflammatory bowel disease [33].

The model in Figure 3 is further supported by studies in CASP8 knockout mice [34]. The caspase 8 deficiency prevents RIPK1 induced apoptosis but allows RIPK3 induced necroptosis (Figure 1B). The lack of restraint results in defective clonal expansion of thymocytes as necroptotic cell death is unchecked during positive selection [34]. In dual RIPK3 and CASP8 knockout mice, the phenotype is rescued as neither of the ZBP1-dependent cell death pathways is active within the thymus. Instead, the CASP8, RIPK3 deficient mice develop a lymphoproliferative phenotype, similar to that found in some models of autoimmune disease. In these mice, autoimmunity is limited as self-reactive thymocytes undergo elimination via external cell death pathways (ECD), such as those activated by FAS ligands [35]. Under these conditions, neither RIPK3 or ZBP1 is essential for negative selection, explaining the lack of an autoimmune phenotype with either ZBP1 or RIPK3 deficient animals [36,37,38]. Collectively, the studies demonstrate that RIPK3 is the sole regulator of necroptosis. The absence of redundant activators of this pathway is consistent with its recent evolutionary elaboration [39,40,41].

## 6. ZBP1 and Mendelian Disease

ZBP1 likely performs similar roles in some non-lymphoid tissues (Figure 1C). In melanocytes, ZBP1 is potentially activated by the DNA damaging effects of sunlight and skin-penetrant environmental mutagens to induce either RCD or ICD [42,43]. ZBP1 may play a role as a genetic modifier in some human mendelian diseases. For example, in Dyschromatosis Symmetrica Hereditaria, pigmentation loss in the skin may follow from ZBP1-induced melanocyte cell death, with the threshold for ZBP1 activation lowered by ADAR1 haploinsufficiency present in this disorder [4]. A similar ZBP1-dependent mechanism is a likely explanation for the pigmentation defects observed in neural crest specific *Adar* knockout mice [44]. ZBP1 may modify pathology in other diseases. In AGS, ADAR1 p150 loss of function Zα variants produces an interferonopathy associated with severe neurological damage [4]. While interferon greatly increases ZBP1 expression, ZBP1 activation is also higher because the ADAR1 loss of function variants results in higher levels of ZNA [29]. Collectively, these observations connect ZNAs and ZBP1 with various disease outcomes and contribute to our understanding of ZBP1 roles in T cells and melanocytes. The findings do not address the functions performed by ZBP1 in the B cell lineage.

## 7. Plasma Cells Are Different from other Immune Cells

Of particular interest is the role of ZBP1 in antibody producing plasma cells (PCs). The basal level of ZBP1 mRNA is higher in PCs compared to T cells, NK cells and other B lineage lymphocytes (Figure 2C, panel 1). One reason may be differences in regulation by interferon regulatory factors (IRF). In PCs, tonic ZBP1 levels are likely regulated by IRF4 [45], which is a PC differentiation factor [46], rather than by those IRFs that drive inflammatory immune responses [47]. PCs also differ in other ways. PCs express ZBP1 but not other nucleic acid sensors such as cGAS (cyclic GMP-AMP synthase), IFIH1 (MDA5), DHX9, DDX41, DDX58 (RIGI) and DHX58 (LGP2) that sense viral threats and drive inflammation (Figure 4B). In contrast, PCs coexpress with ZBP1 the subset of nucleic acid sensors that suppress endogenous retroelements and viruses, such as IFI16 and ADAR (Figure 4B) [48,49,50,51]. One interpretation is that PCs use a different strategy from T cells to protect the host from pathogens.

## 8. Plasma Cells as Factories for Viral Production

PCs differ in important ways from other classes of immune cells. They have a unique biology related to their high levels of immunoglobulin production. Their survival depends on the unfolded protein response (UPR) to minimize the stresses associated with protein aggregation and antibody secretion [42]. With their optimized protein production line, PCs are an ideal factory for viruses to replicate themselves in. 

The Epstein–Barr herpesvirus (EBV) efficiently exploits PCs to produce infectious virions and to maintain transmission to other hosts. Following the initial infection when production of infectious virus first occurs, EBV then enters a latent phase where it produces no virions. The virus takes up residence in memory B cells and syncs its replication with that of the host [53,54]. Between 2 and 500 episomes exist in each infected cell [55]. When memory B cells differentiate into PCs following antigen activation, the virus switches to the lytic phase of its cycle to drive its own replication and produce infectious virions. EBV utilizes the very same transcription factors (TF) that drive PC differentiation to initiate the transition from one replication mode to the other. These TF include POU2AF1 [56], TNFRSF17 (also called B-cell maturation antigen (BCMA)), XBP1, ATF4, KLF13 and MEF2B [45,52] that are coexpressed with ZBP1 (Figure 4A–C). The transcription factors XBP1, PDRM1 and IRF4 are also components of the UPR that is regulated by the heat shock protein HSPA5 [42] and facilitate the packaging of EBV replicons.

## 9. Plasma Cell as Sensors of Tissue Pathology

The heat shock proteins that PCs express at high levels serve as DAMPs to warn of the threat when viral replication overburdens the UPR. Virally encoded proteins also serve as PAMPs, compensating for any defects in the presentation of viral peptides to adaptive immune cells by major histocompatibility antigens. TRCs instead provide the first line of defense.

The DAMP and PAMP pathways enable PCs to signal their distress. PCs can act as whole cell sensors to detect metabolic and oxidative stress in their local environment. They integrate the threat level in real time and signal TRCs to initiate anti-viral responses by executing the nuclear option: ICD. A similar role for PCs as sensors of neighborhood dysfunction likely explains the association between improved survival of patients when PCs are present in tumors [57] and cases of TNF-resistant inflammatory Crohn’s disease due to PC infiltrates in the affected tissue [58]. In each case, ICD initiated by PCs perpetuates the inflammatory cycles by promoting the differentiation of responding cells into PCs and TRCs.

## 10. ZBP1 and Viral Retorts

So what role is ZBP1 performing in PCs? ZBP1 could protect the host by initiating PC death, acting as a sensor for the large amounts of ZNAs produced during the transcription, replication, processing and packaging of viral genomes. If this were so, then it would be expected that viral counter measures would enable evasion of the ZBP1-dependent host restriction factors. Indeed, viruses such as vaccinia virus [59], cytomegalovirus [60] and herpes simplex [61] prevent activation of RCD and ICD by encoding proteins that prevent activation of RIPK1 and RIPK3 by ZBP1. However, EBV is not known to use this strategy.

EBV has evolved other schemes to block cell death, including inhibition of apoptosis by latent membrane proteins (LMP), and small RNAs that target RIPK1/RIPK3 dependent pathways and not ZBP1 [62,63]. Additional EBV encoded genes suppress externally activated cell death pathways by blocking the expression of immunogenic viral genes on the PC surface [64,65,66]. While EBV has many ways to ensure the survival of infected PCs, blocking ZBP1 is not one of them.

So, if the ZBP1 expressed within PCs has no impact on the elimination of virally infected cells, why is basal expression of ZBP1 so high? We propose ZBP1 plays an important role in enforcing EBV latency to prevent EBV lytic replication. While this outcome results in persistent infection of B cells by EBV, it minimizes the serious health threat posed to the host by the virus.

### Mono a Mano

EBV, the “kissing virus”, is the cause of infectious mononucleosis (IM, also known as glandular fever) and infects almost everyone. Once transmitted, the virus persists for life. EBV infection is a significant cause of morbidity and mortality. About 1–1.5% of worldwide cancer incidence are linked to the virus [19,21]. EBV is also associated with an increased risk of autoimmune diseases, such as multiple sclerosis [20] systemic lupus erythematosus (SLE), although often the connections have been hard to prove using epidemiological approaches given the high rate of seropositivity in the population [18,67]. We hypothesize that there is benefit to both host and virus if the role of ZBP1 in PCs is to promote EBV latency. The coadaptation increases the likelihood that both will survive to propagate the next generation. Being “frenemies” is the best option for both. The host and virus know each other well, but clearly do not do anything to give the other an advantage.

## 11. EBV and Its Propensity to form Z-DNA

The “mono a mano” struggle between the virus and host should be reflected in the evolution of EBV and host genomes. For example, there should be selection against ZNA sequences in the viral genome that are capable of activating ZBP1-dependent restriction. One way of assessing this possibility is by measuring the frequency of Z-DNA forming sequences in EBV relative to the other herpes viruses that do not infect B cells. The Z-forming potential of EBV can be scored computationally [2] and is strongly related to the GC content of a genome. Since Z-formation is based on a dinucleotide repeat with an alternating *syn* and *anti* orientation of nucleotide bases relative to the ribose ring [68], the expectation is that the number of Z-DNA forming sequences should increase with the GC content of a genome (in proportion to the probability of finding consecutive GC repeats), as indicated in Figure 5B by a dotted line. However, for both EBV and Kaposi virus (HHV8), which is also B-cell tropic [69], the frequency of Z-DNA forming sequences falls below this line, indicating negative selection against Z-DNA in EBV and HHV8 genomes relative to other herpes viruses (Figure 5B). The remaining strong Z-DNA forming sequences (as indicated by the vertical lines in the red box in Figure 5A) are found in promoter regions, suggesting that they regulate gene expression of the virus. Of particular interest is the EBNA-LP gene that is involved in establishing EBV latency. Positive selection of EBNA-LP is indicated by the almost total conservation across strains of the repeat sequences within it, including the ZNA forming elements, a feature not present elsewhere in the viral genome [70].

There is also evidence of host adaptations that could reflect a role for ZBP1 in EBV and other persistent viral infections. In particular, ZBP1 is under positive selection in primates, although the rate is less than that for RIPK3 and MLKL [39,78]. Further evidence of selection is provided by the existence of over 2000 possible ZBP1 splicing isoforms. The diverse transcripts may have enabled humans to escape post-transcriptional suppression by the type of EBV1 noncoding RNAs (ncRNAs) that inactivate other host anti-viral responses [79]. The challenge ZBP1 poses to the virus is whether the virus can express sufficient ncRNAs at high enough levels to inactivate all possible host isoforms. The advantage here definitely belongs to the host as many of the ZBP1 splice isoforms are non-functional and act as decoys for the virally produced ncRNAs. In this scenario, ZBP1 protein would normally silence host decoy mRNA isoform production via a negative feedback loop, while viral suppression of ZBP1 translation would increase decoy output. Of course, it is not possible to attribute all such genetic scars to the battle with EBV.

## 12. ZBP1 and the Silencing of EBV

So, what function does ZBP1 perform in PCs that other anti-viral sensors may not? We note that ZNA-forming sequences are able to modulate gene expression in some experimental models, with effects correlated with Z-DNA formation [80,81,82]. The most common way to induce the flip from right to left-handed DNA is through the action of an RNA polymerase, which powers the formation of Z-DNA as it separates the DNA strands to make RNA. The negative supercoiling produced by underwinding the right-handed B-DNA helix is relieved by the reverse twist of the left-handed Z-DNA helix formed upstream of the polymerase [83]. The forward and reverse rates for the B-Z transition are in the millisecond range for the best Z-DNA forming sequences, illustrating the highly dynamic nature of the process [84]. The flip to Z-DNA can also occur during DNA replication. Z-DNA also forms during the paranemic pairing produced by strand exchange during homologous recombination of topological closed domains [85] and in form V DNA [86,87]. In addition, the Z-DNA flip is driven by the eviction of nucleosomes from chromatin. The ~1.65 DNA turns released by uncoiling DNA from the histone octamer is sufficient to flip one helical turn of B-DNA to Z-DNA [84,88]. The equivalence means that within any loop, there is ample energy stored in nucleosomes to power Z-DNA formation

## 13. Flipons and EBV Gene Expression

The sequences that flip to Z-DNA or Z-RNA under physiological conditions, named flipons, create binary switches where the DNA helix is either right- or left-handed, providing building blocks for a digital genome where responses are context specific and depend on the flipon conformation [71]. The flipons act by recruiting different sets of cellular machinery to DNA and RNA depending on its handedness, varying the readout of genetic information through effects on transcript expression, editing and splicing. As flipons impact phenotypes, they are subject to natural selection [71]. Flipons are often encoded by simple sequence repeats that can also contribute to condensate formation by coding for intrinsically disordered peptides [88]. Given the high frequency of ZNA forming flipons in the human genome, it is unlikely that any two cells share the same overall DNA conformational state, with the flipon settings optimal for survival varying by context [88].

The positive selection of Z-DNA forming elements in genomes provides evidence that they operate as flipons and enables insight into their function. The high conservation of a Z-RNA forming sequence block (Z-box) in the human *Alu* retroelement provides one example of this outcome. The Z-box localizes ADAR1 p150 to host RNAs and prevents the activation of anti-viral interferon responses against self [89]. The enrichment of Z-DNA forming elements in human and mouse promoters [83,90], as well as those of EBV, is also consistent with a role for Z-DNA in regulating gene expression.

## 14. Flipons and EBV Latency

We propose that ZBP1 acts on EBV flipons to maintain viral latency. The outcome benefits both host and virus. The compromise protects the host from pathologies arising from EBV lytic replication while favoring the virus by ensuring its persistence. How then might ZBP1 maintain EBV latency? Three different but not mutually exclusive mechanisms are presented in Figure 6, with the proteins involved listed in Figure 6A. The first two possibilities are based on well documented pathways that induce the suppressive dimethylation of the histone H3 lysine 9 residue (H3K9me2), initiated either by the MYC oncogene protein or nuclear factor kappa beta (NF-κB) (Figure 6B,C). The other pathway depends on the highly conserved EBV encoded EBNA-LP gene (Figure 5A and Figure 6D).

## 15. The FACT of MYC Induced Latency

In vitro assays confirm that EBV latency depends on the host encoded MYC and the FACT (Facilitates Chromatin Transcription) chromatin remodeler that is composed of SSRP1 (structure specific recognition protein 1) and SUPT16H (SPT16 homolog) subunits [66,72,91,92,93]. MYC works with FACT to suppress viral gene transcription. By increasing expression of each other, MYC and FACT collectively create a positive feedback loop that establishes and maintains EBV latency [94]. Disruption of either MYC or the FACT subunit SSRP1 prevents EBV latency [72]. Activation of EBV lytic program ensues instead [94].

An important role of FACT in normal cells is to suppress transcription of noncanonical promoters. FACT initiated pathways edit nucleosome composition, rewrite histone modifications and promote DNA methylation [95]. FACT is especially important for silencing loci that produce dsRNAs and Z-RNAs that otherwise would activate anti-viral responses within the cell [95,96].

## 16. FACT and Z-DNA

FACT acts by removing histone subunits from DNA [97], a process that can potentially power Z-DNA formation. Indeed, disrupting the interaction of FACT with DNA via the small molecule CBL0137 unmasks Z-DNA in normal cells, indicating that FACT and Z-DNA formation are intimately connected [98]. It is reasonable to propose that the actions of MYC and the FACT subunit SSRP1 are also associated with Z-DNA formation in EBV promoters (Figure 5A), localizing ZBP1 and other Z-DNA binding proteins to these sites. The effect on EBV gene expression would then depend on the chromatin modification complexes (CMCs) these proteins associate with.

One CMC that interacts with ZBP1 is the C-terminal binding protein (CTBP) corepressor that is an important mediator of gene silencing [99] and identified as a host factor regulating B-cell transformation by EBV [100]. ZBP1 co-immunoprecipitates with CTBP1, CTBP2, KDM1A (LSD1) and the zinc finger containing protein ZFN516 [74,101,102]), all components of CTBP. CTPB also incorporates the euchromatic histone lysine methyltransferases 1 and 2 (encoded by EHTM1 and EHTM2) [103] that induce gene silencing by H3K9 methylation, as well as by other histone modifying enzymes. These observations connect ZBP1 to the induction of EBV latency.

## 17. Z-DNA, E-Boxes and H3K9 Methylation

MYC activates the CTBP associated methylase EHMT2 to suppress gene expression [75]. In the case of EBV, the interaction likely occurs after ZBP1 localizes CTBP to the MYC binding sites at which FACT initiates Z-DNA formation (Figure 6B). MYC recognizes enhancer-box sequences (E-box), which are based on a variant of the prototypical CACGTG motif, an alternating purine-pyrimidine motif of the kind that favors Z-formation [104]. E-boxes are also recognized by the CTBP associated proteins ZEB1 and ZEB2 (zinc finger E-box binding homeobox) through their zinc finger domains [103,105,106], resulting in competition of MYC and CTBP for the same binding site. FACT and other chromatin remodelers may provide a mechanism to dislodge one factor so the other can bind. The exchange is potentially driven by the Z-DNA formed when FACT ejects histones from DNA by FACT [83,90]. The energy released by uncoiling DNA from around the nucleosome is captured by Z-DNA formation and remains available to power assembly of incoming protein complexes. In the process, one E-box binding factor is replaced by another. Z-DNA acts as a capacitor, storing energy to catalyzes the exchange (see graphical abstract). Further experiments to validate this model are required.

The localization of CTBP by ZBP1 to flipons within promoters bound by MYC then could lead to the establishment and maintenance of EBV latency by inducing H3K9me2 that then is read by other CMCs to write or erase other epigenetic marks that affect gene expression. Engagement of different classes of E-box binding proteins at sites of Z-formation could also produce other outcomes [107]. For example, engagement of the EBV encoded BZLF1 protein would induce EBV lytic replication rather than latency [108], while binding of the BRG1 subunit the SWI/SNF (encoded by SMARCA4) to sites of Z-DNA formation would promote chromatin remodeling [81,82,109,110], impacting the transcription of viral genes. Outcomes then reflect the regulation and the relative affinity of the different CMC competing with each other at locations where Z-DNA forms and are modulated by the factors already engaged at those sites. In these situations, Z-DNA only seeds a CMC condensate. It does not specify what happens next.

## 18. NF-κB, Z-RNA and H3K9 Methylation

The other mechanism of Z-dependent silencing in EBV promoters involves activation of NF-κB by RIPK1. The pathway depends on the ZNAs produced during the early stages of EBV infection, with ZBP1 activating RIPK1 to promote ubiquitination of IκBα and release of p50 and p65 (encoded by NFKBIA, NFKB1and RELA, respectively) (Figure 1B and Figure 6C). The pathway is active before latency is fully established, a process that can take several days, providing a possible window for therapeutic intervention [77]. While NF-κB is normally regarded as pro-inflammatory, silencing of interferon-dependent genes by p50 homodimers is well described [73] and likely limits induction of interferon-β by p65 [111]. The p50 homodimer lacks a transactivation domain and is present in the nucleus of resting cells [73]. The homodimer binds to NF-κB response elements in DNA. Rather than inducing gene activation, p50 promotes H3K9 methylation to silence gene expression. During chronic infection, p52 homodimers may perform a similar role.

Gene repression depends on EHMT1 rather than EHTM2, which is engaged by MYC. In both cases, the subsequent modifications by EZH2 (enhancer of zeste 2 polycomb repressive complex 2 subunit) result in H3K9me3 and H3K27 trimethylation [75]. Such effects would promote latency of EBV genes with NF-κB binding sites in their regulatory regions. 

The effect of EHMT and EZH2 inhibitors on this process depend on the stage of infection. Paradoxically, administering the drugs at the time of primary exposure unexpectedly suppresses EBV replication rather than increasing expression of viral genes to enhance virulence. In this situation, the drugs prevent silencing of the host anti-viral genes by p50 induced histone methylation, allowing for efficient elimination of the virus before it can reprogram the host cell [73,112].

## 19. EBNA-LP EBV Gene Silencing and a Role for Flipons

The third mechanism is similar in principle to the way *Xist* silences genes [113] (Figure 6D,E). The outcome depends on the latency inducing gene EBNA-LP. This gene contains multiple copies of two non-overlapping repeat elements: a sequence that forms Z-RNA along with a stable intron sequence that folds into a 586 base dsRNA hairpin (sisRNA) [64]. The EBNA-LP gene also contains multiple copies of the W1 and W2 exons that together encode an intrinsically disordered peptide composed of many basic amino acid repeats [114] typical of those found in RNA binding proteins [115]. The peptide potentially engages with sisRNA to form a scaffold, seeding formation of a condensate capable of silencing EBV lytic gene expression much as the human *Xist* RNA inactivates one copy of the X-chromosome in females. In addition to ZBP1 and the complexes associated with it, the scaffold could engage additional ZNA binding proteins, such as the dsRNA editing enzyme ADAR1 or the many ZNA binders still awaiting discovery. Indeed, the reported adenosine to inosine substitution found in sisRNA is consistent with docking of ADAR1 to the sisRNA scaffold [116], although the significance of the observed editing event at this site and in the BHLF1 mRNA [117] is currently unknown.

The highly conserved nature of the EBNA-LP repeat sequence elements (LPR) across strains (Figure 6D) provides evidence that they are subject to positive selection [70]. In contrast to the sequence conservation, the number of LPR copies present in the EBNA-LP gene of each strain is variable, ranging from two to nine copies. At least five LPR are necessary for optimal infection of B cells [118]. Each LPR contains an alternative transcription start site and can promote differential splicing of the EBNA-LP pre-mRNA, leading to transcripts of different length (Figure 6D). The longest transcript is the major one present in established infections [77,119]. Alternative RNA processing can also produce bicistronic transcripts that incorporate exons from other EBV genes. Promoter usage can vary as well, leading to out-of-frame EBNA-LP RNAs that are non-protein coding. Potentially these transcripts could function as a ncRNA [120] that modulate both host and viral responses [66].

The variable length of EBNA-LP RNAs represents a way to match EBV latency to the cellular context. For example, a particular length transcript could be optimal for tuning latency at each stage of B-cell development. Z-formation by the EBNA-LP DNA repeat elements would be expected to modulate this process. Flipping an element to Z-DNA would pause transcription by both upstream and downstream RNA polymerases, providing time for the splicing machinery to assemble on the transcript to change the isoform produced [121]. The mechanisms involved are all experimentally addressable with existing assays.

## 20. EBV Latency and Potential Flipon Induced Changes to EBV Genome Topology

Other topological effects of the flip to Z-DNA depend on the circularization of the EBV episome by proteins that approximate the genomic ends to each other. The formation of Z-DNA within this DNA ring would alter the episomal topology, changing the contacts made between different regions of the EBV genome by the bending the DNA at B-DNA/Z-DNA and Z-DNA/Z-DNA junctions (~11° and ~25°, respectively) [122,123]. The sub-genomic loops formed then depend on which segments adopt the Z_DNA conformation.

Loops may contain a number of flipons that vary in length and sequence composition. The different flipons can compete against each other for the available torsional energy to affect gene expression. Better Z-forming sequences will flip first at low levels of negative supercoiling [124], with other flipons remaining in the B-DNA conformation. As the level of negative supercoiling rises, other flipons will initiate Z-DNA formation. If the second sequence to flip is long enough, it will absorb all the available torsional energy and revert the first sequence back to B-DNA. This effect is purely topological and can alter the conformation of sequences separated by large distances, even when the DNA in between is protein bound [124]. The competition between Z-DNA forming elements provides a mechanism to regulate gene expression over many megabases in the host genome. A change in flipon conformation at one locus then alters flipon conformation at another, with effects on gene expression and repression. Flipons based on simple sequence repeats of variable length can then impact phenotype even when their alleles are not in linkage disequilibrium either with the local or distant SNPs used in genome wide association studies for the mapping of human trait variation to genes [121,125]. Flipon genetics translates repeat variation into genome topology.

The effects of competition between flipons should also be apparent in small genomes like EBV. Indeed, the three-dimensional topology of EBV varies according to the type of latent infection as classified by the number of genes expressed [126,127]. There is no viral RNA in type 0 latency which is found only in infection of non-dividing memory B cells. In type I latency, EBNA1 is the only protein expressed and enables the virus to replicate in sync with the host. In type II latency, all latency genes except EBNA 2 and EBNA3 are expressed. In type III latency, all latency genes are transcribed [128]. On initial infection of naïve B cells, type III latency is established with progressive restriction of EBV gene expression as the infection continues and immunity to EBV gene products develops [126]. While CTCF and cohesion are major determinants of host chromatin organization leading to differential gene expression, their contacts with EBV DNA do not differentiate between different latency states, suggesting other factors are involved in setting viral topology during latency [127]. In contrast, loop formation during infection varies with MYC expression [72], raising the question of whether the changes in flipon conformation induced by MYC and FACT play a role in how the EBV genome folds and which viral genes are expressed.

## 21. The EBV Lytic Program Depends on Fully Methylated DNA

The epigenetic silencing of EBV lytic genes also involves DNA methylation, which is not only essential for maintaining latency but also provides the virus with an exploit for activating the lytic program at some later time. The lytic switch is mediated by the EBV BZLF1 protein that only binds with high affinity to fully methylated EBV DNA [129,130]. This requirement requires an additional layer of regulation that involves the EBV encoded EBNA2 protein [131]. EBNA2 acts with EBNA-LP to increase the enzymatic oxidation of the DBA base 5-methylcytosine by the family of 5-methylcytidine TET methylcytosine dioxygenases [129,130]. The enzymes catalyze the production of 5-hydroxy- and 5-carboxy cytosine from 5-methylcytosine. These adducts prevent the docking of BZLF1 to its target sites. The adducts also promote Z-DNA formation, particularly in partially methylated flipons [132], providing a way to localize the cellular machinery to these regions. These CMCs ensure the complete removal of adducts from the EBV DNA and that the genome is fully methylated. The back and forward nature of such reactions sets a threshold for initiating the viral lytic program based on the state of DNA modification. The level of EBNA2 expression and the nature of the EBNA-LP gene products are then two factors that influence when the lytic switch is thrown [131]. The levels of both factors vary with context, preventing lytic replication when their expression is high.

## 22. Stopping EBV Lytic Replication

A number of feedback mechanisms depend upon the CTBP complex to protect the host against the onset of EBV virus lytic replication. CTBP is regulated by HIPK2 (homeodomain interacting protein kinase 2), a key element of the integrated stress response. HIPK2 provides a means to sense viral replication, likely triggered by failure of the UPR. HIPK2 destabilizes CTBP, activating apoptosis in both a P53 dependent and P53 independent manner, both pathways capable of closing down the viral production factory [133].

CTBP effects are also regulated by the cell death initiator poly(ADP-ribose) polymerase 1 (PARP1) [134], which is highly expressed in PCs (Figure 4B). Both CTBP and PARP1 use nicotinamide adenine dinucleotide (NAD) as a cofactor, with CTBP activity increased by NADH [135,136], while NAD^+^ is a substrate for PARP1. Depletion of cellular NAD following PARP1 activation during times of metabolic stress reduces NADH levels and favors both viral replication and lytic gene activation by repressing EZH2 expression, thereby reducing H3K9me3 and H3K27me3 levels. PARP1 modifications also activate pro-inflammatory NF-κB pathways [126,137,138]. In other contexts, PARP1 along with tankyrase limits EBV virulence by PARylation of EBNA1 to suppress EBV OriP replication [139] and by inducing cell death through either necrosis or apoptosis [140].

ZBP1 and other Z-DNA binding proteins may further protect the host by modulating expression of cell genes. For example, MYC expression has Z-DNA forming flipons in its promoter that could affect its expression. ZNA binding proteins that increase MYC transcription could then reinforce viral latency, while increasing the cell’s resilience to replication stress [96,141,142,143]. Similar processes could modulate other host genes important to EBV for its maintenance and survival. The complex cellular circuitry involved is the result of a long period of coadaptation between host and virus, ensuring that the most likely outcome of any particular encounter is a stalemate. The virus then persists only by replicating in sync with the host.

The flipon-dependent outcomes we propose offer novel insights into the biology of chronic viral persistence. The host tolerates the virus while minimizing negative outcomes from persistent infection. Like the version of the prisoner’s dilemma that reiterates indefinitely, the optimal outcome for both partners is to cooperate [144]. Of course, as in any game, either EBV or the host can defect, optimizing outcomes that provide them with an immediate advantage.

## 23. A Model for Disease When EBV Defects

Escape of EBV from latency leads to lytic replication and spread of the virus throughout the population. There are also many adverse outcomes for the host when the virus prevails, including death during the acute phase of infection [145]. In survivors, somewhere between 1–40% of the total CD8^+^ T-cell pool from IM patients is reactive to EBV epitopes, usually from proteins associated with lysis [53]. There is a risk that autoimmunity will develop through cross-reactivity with self-antigens [146,147]. Some antibodies that develop are even specific for Z-DNA [148], which can be presented to B cells by bacterial biofilms. An increase in levels of anti-DNA antibodies produced (only some of which recognize Z-DNA) is a well-established biomarker for disease flares in SLE [149]. The plasma cell differentiation induced by bacterial DNA potentially also activates EBV lytic replication [150,151].

With repeated cycles of latency and lysis, or when repression of EBV genes is leaky, depletion of antigen-specific T cells [31] may result. The holes created in the immune repertoire likely include those high affinity, viral-specific clones needed to defend against tumors. Chronic inflammation would in this situation promote tumor escape from immune surveillance by further restricting the clonotypic diversity of the available T cell repertoire [152]. EBV induced suppression of host tumor suppressor genes may also promote malignancy, leading to tumors lacking an inflammatory infiltrate when the expression of immunogenic, tumor-specific EBV proteins is also silenced (Figure 7).

## 24. Unhosting EBV

Therapies aimed at plasma cells, where EBV undergoes lytic replication, have been reported as successful in multiple myeloma [153] and SLE [154,155] but have significant toxicities. Other therapeutic approaches are possible. Strategies specifically targeting the highly conserved EBNA-LP transcript should reverse many of the tactics employed by EBV to hide its immunogenic self from the immune system. The combination of anti-viral therapeutics with inhibitors of H3K9 or H3K27 methylation that reverse latency may be synergistic and enhance the killing of infected cells while preventing viral escape by lytic replication. Treatment of latent EBV infections by this approach may be optimal during disease flares when differentiation of plasma cells enables viral lytic replication, coinciding with an increase in anti-dsDNA levels [149]. Over time, the immune response that follows may completely eliminate replication competent virus from patients [112]. Alternatively, treatment with inhibitors of H3K9 or H3K27 methylation at the time when individuals first present with infectious mononucleosis may prevent the establishment of persistent EBV infection. Administration of drugs targeted at epigenetic modifications would be most effective in the time period before latency is established [73]. Another cost-effective strategy is implementation of a childhood vaccination program to decrease the rates of EBV infection in the population.

## 25. Future Directions

The study of the immune system has often led to the discovery of new biology, including biological roles for Z-DNA and Z-RNA. Further insight into how alternative DNA structures can alter gene expression is possible using the EBV latency model. The virus provides an experimentally tractable system for unraveling the connections between ZNA, CTBP, heterochromatin, the integrated stress response and metabolism. The importance of this work is underlined by the unmet needs in autoimmune diseases and cancers due to persistent EBV infection. The therapeutic approaches arising from these explorations extend to other chronic viral infections and diseases where flipons play an essential role in their etiology.

## Figures and Tables

**Figure 3 ijms-23-03079-f003:**
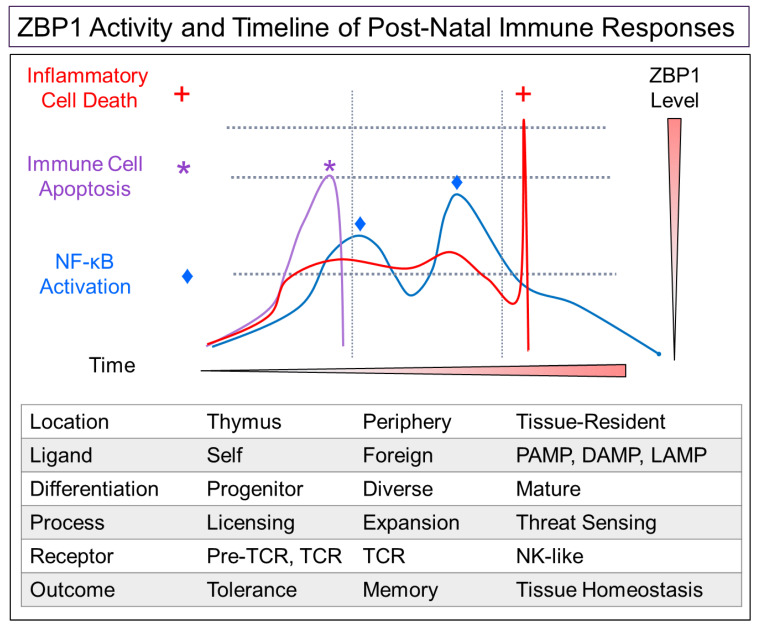
A proposed timeline for immune cell development highlighting different roles for ZBP1 in protecting against threats. Early in development, ZBP1 is able to protect against reactivation of endogenous retroelements and aberrant regulation of transcription by sensing the dsRNA involved and inducing apoptosis (indicated by *****). Late in development, ZBP1 expressed by tissue-resident immune cells can activate necroptosis (indicated by **+**) when pathogen associated molecular patterns (PAMPs), damage associated molecular patterns (DAMPs) or live-style associated molecular patterns (LAMPs) are detected. The inflammatory cell death clears space for responding T cells to eliminate the threat. NF-κB is activated by ZBP1 during an adaptive immune response (shown by ♦).

**Figure 4 ijms-23-03079-f004:**
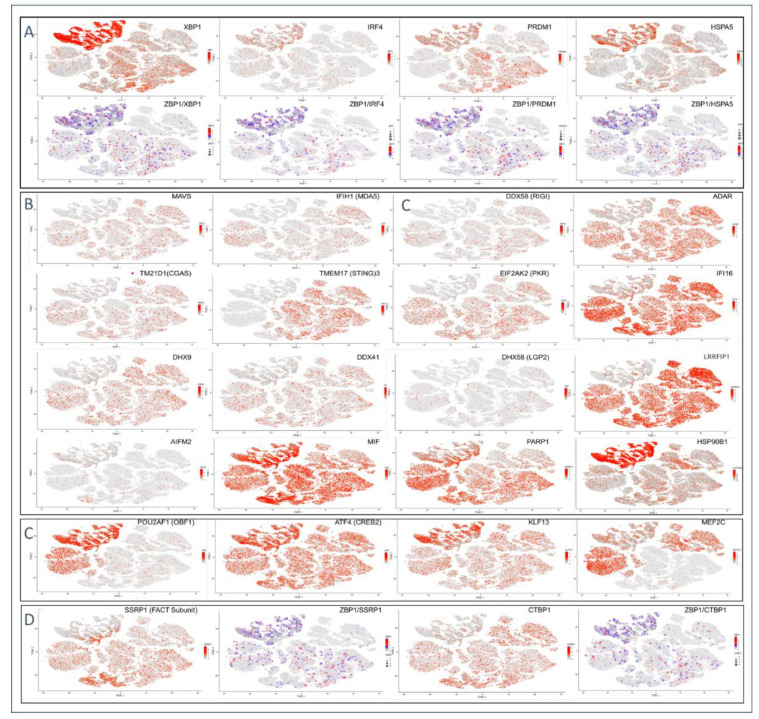
(**A**). Plasma cells have high expression of unfolded protein response (UPR) genes including key transcriptional factors XBP1 (X-box binding protein 1), IRF4 (interferon regulatory factor 4) and PRDM1 (PR/SET domain 1), along with the regulator HSPA5 (heat shock protein family A (Hsp70) member 5) as shown in the upper panel. The middle panel confirms coexpression with ZBP1 with the smallest dots representing cells that lack ZBP1 expression. (**B**). Expression of RNA and DNA nucleic acid sensors and their effectors show no or low coexpression in plasma cells relative to other immune cells from blood, liver and spleen (upper two rows). The expression of apoptosis inducing factor (AIFM2) is also low in plasma cells, but the effector proteins PARP1 (poly(ADP-ribose) polymerase 1) and MIF (Macrophage Inhibitory Factor) of the ADP-dependent apoptosis pathway are robustly expressed along with the heat shock 90 protein HASP90B1. (**C**). Transcription factors expressed in plasma cells belonging to gene families with proposed roles EBV gene expression [45,52]. (**D**). Coexpression of ZBP1 with the FACT (Facilitates Chromatin Transcription) component SSRP1 (structure specific recognition protein 1) and carboxy terminal binding protein 1 (CTBP1).

**Figure 5 ijms-23-03079-f005:**
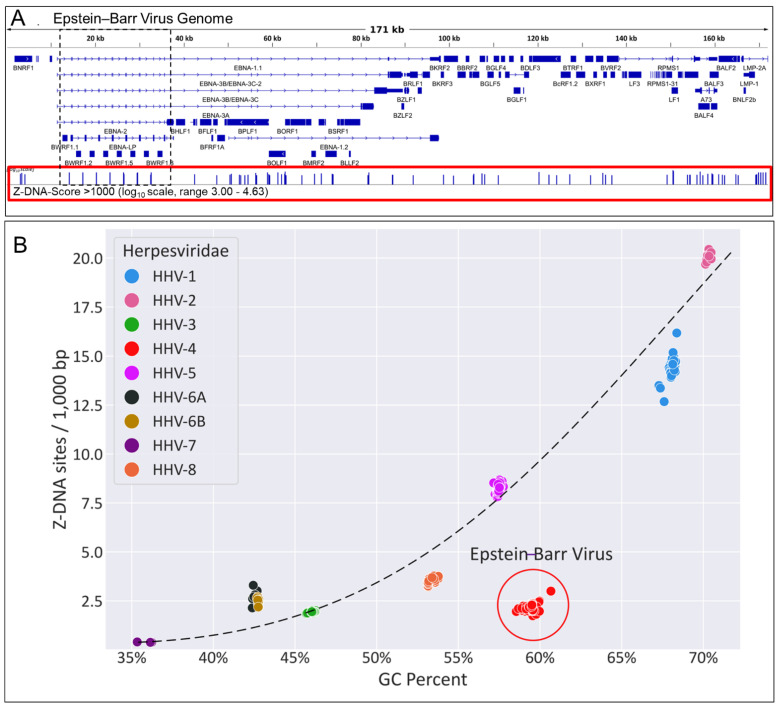
ZBP1, flipons and disease with evidence of the evolutionary selection against Z-DNA forming regions in the EBV genome. (**A**). The EBV genome is characterized by strong Z-DNA forming segments in gene promoters. Vertical lines in the red box show the position of sequences with a Z-Score greater than 1000, as determined using the Z-HUNT3 algorithm [2]. Sequences such as these that change their DNA conformation under physiological conditions are called flipons. They can act as switches to turn gene expression “on” or “off” [71]. The region in the dotted box is expanded on panel D of Figure 6 (**B**). A plot of GC content of different herpes virus (HHV) genome sequences against the number of sequences per 100,000 base pairs with a high propensity to form Z-DNA under physiological conditions (with ZHUNT3 scores >1000). Independent isolates from the various labeled strains are shown. The dashed line represents the expectation that the number of Z-forming sequences increases with GC content. The Epstein virus group of sequences is shifted to the right of this line, consistent with selection against Z-DNA forming sequences in plasma cells.

**Figure 6 ijms-23-03079-f006:**
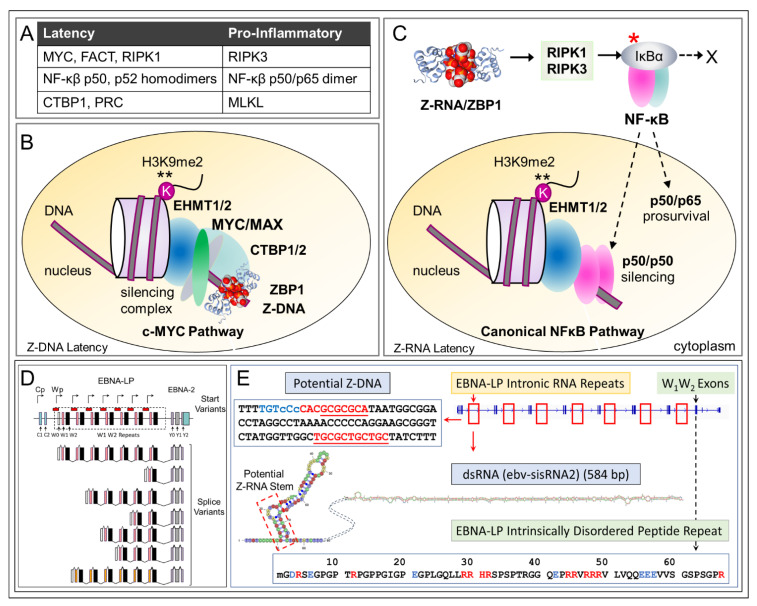
Different pathways for ZNA dependent EBV latency. (**A**). The factors and pathways associated with ZNA latency differ from those shown in Figure 1 that promote ZBP1 induced cell death. The MYC pathways depend on the interaction with the FACT complex as described in the text [72], while the NF-κB dependent transcriptional silencing likely rely on the p50 homodimer in acute infection and the p52 homodimer in chronic infection. Both dimers lack a transactivation domain and are known to induce histone H3K9 methylation [73]. (**B**). Z-DNA dependent latency. The MYC induced Z-DNA formation localizes ZBP1 and other members of the CTBP1 complex to that region [74]. MYC also binds EHTM1 to induce H3K9 methylation [75]. (**C**). Z-RNA induced latency. The outcome of NF-κB activation depends on context. In addition to well-known pro-inflammatory roles [76], the p50 homodimer (encoded by NFKB1) can induce H3K9 methylation to suppress interferon induced genes [73]. In this case, phosphorylation of the IκBα (indicated by a red asterisk) protein leads to its ubiquitination and removal by proteolysis (indicated by a cross). The p50 and p65 NF-κB subunits can then enter the nucleus. The non-canonical NF-κB pathway is based on p52 homodimers encoded by NFKB2. (**D**). The EBV early region contains the essential latency protein EBNA-LP gene (See Figure 5A for entire EBV genome) and repeats of the W1 and W2 exons, as indicated by differently colored vertical stripes. W0, C1, C2 and Wp represent EBV promoters. Alternative transcription start sites are indicated by the bent arrows. RNA splice forms are also presented (based on Figure 1 from [77]). The red boxes indicate the intron repeat sequences that overlap the BWRF1 open reading frame and encode 586 base pair RNA hairpins [64]. (**E**). The potential EBNA-LP scaffold for regulating EBV gene expression consists of an intronic long-noncoding RNA that forms Z-RNA and RNA hairpins, plus the intrinsically disordered peptide the gene encodes (basic amino acids are colored in red). The Z-DNA forming sequences in the 7 repeats are indicated in red letters with blue indicating an extended Z- forming region that lacks a perfectly alternating purine/pyrimidine repeat (indicated by dotted lines under residues and small letters for the bases out of alternation). The Z-RNA stem (shown within the dotted box) forms from the complementary bases that are underlined. The noncoding RNA and peptide have the potential to nucleate a condensate that functions like *Xist* to silence the EBV genome.

**Figure 7 ijms-23-03079-f007:**
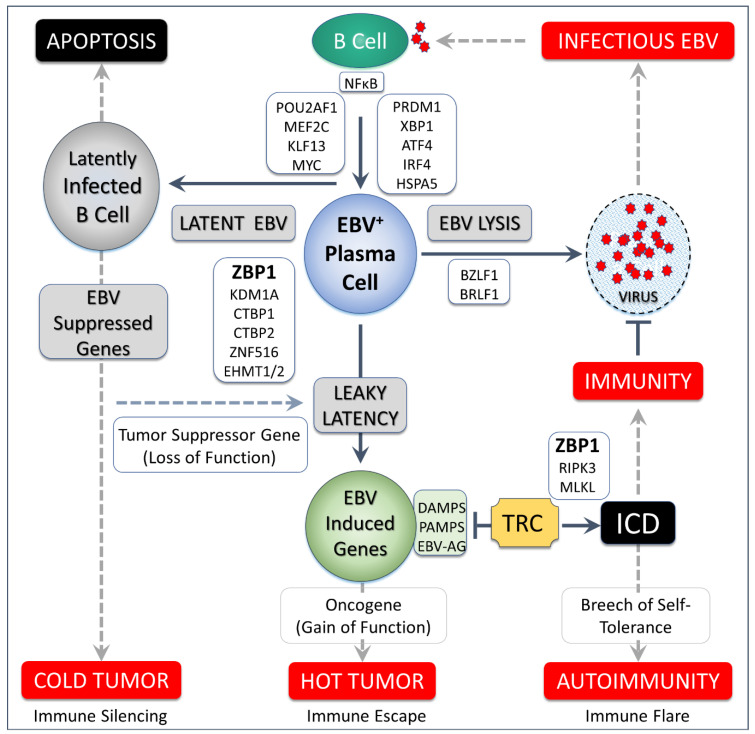
Epstein–Barr virus persists as a latent infection in B cells. Plasma cell formation can trigger the lytic program with three possible outcomes. The lytic program may complete and produce virus that spreads the infection. Latency may be maintained. Alternatively, suppression of EBV genes may be leaky with variable and partial suppression of viral genes, producing DAMPs (damage associated molecular recognition patterns), PAMPs (pathogen associated molecular recognition patterns) and EBV antigens (EBV-AG) to activate tissue resident cells (TRC) that act as an early warning system and induce inflammatory cell death (ICD). While TRCs may induce immunity, over time autoimmunity may develop. TRCs may also serve as inflammatory drivers of tumorigenesis. White boxes show host encoded transcription factors expressed in plasma cells, some of which regulate activation of EBV lysis [72]. ZBP1 is reported to bind components of the CTBP transcriptional corepressor complex [74] and may target it to Z-DNA formed in EBV promoter regions to maintain latency. EBV induced suppression of tumor suppressor genes and activation of oncogenes, including those encoded by the virus underlie the ~1-1.5% of cancers caused worldwide by the virus.

## Data Availability

Publicly available data were analyzed as described in the legend for Figure 2 using the R-project statistical language. The code is available from the authors on reasonable request.

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
