# Peer review of "Mono a Mano: ZBP1’s Love–Hate Relationship with the Kissing Virus"

_ijms, 2022, doi:10.3390/ijms23063079_

Round 1

Reviewer 1 Report

Manuscript by Herbert et al summarizes data on Z-DNA, ZBP1, ADAR involvement, etc. Text is generally well-written and thoughtful. Based on background knowledge plus some new bioinformatic findings authors propose new model on ZBP-1 - HBV intercations which might have future therapeutic value. The idea is clearly interesting, is supported by the background and computational findings; whether it is really the case and will it work in a therapeutic setting, only more research will tell. Moreover, some of the authors views/terminology are somewhat controversial (e.g., what is meant by "digital genome" in the article's context?). For the review/insight this is nevertheless OK (maybe manuscript would better fit into "Opinion" or similar article type though), however I do have a few concerns which has to be addressed.

1) This is submitted as a review, however some new data is presented. This is fine but has to be explained. I.e. what was done specifically and how. Otherwise there is no way to reproduce etc.

2) Lane 23 (and later in the manuscript) "Our focus is on Epstein-Barr virus (EBV), a24 pathogen that causes significant human morbidity and mortality." - This is an overstatement..
Also for this, lane 208: "About 1-1.5% of worldwide cancer incidence are linked to the virus." - where does this statistics comes from?? 1 percent is a huge number.

3) Figures organization need improvements. For example, authors present their data on Fig. 2 and then jump back to Fig. 1 to present their model. This disrupts the flow of the paper. Fig. 4 has similar issues. Need to align text to figs throughout the manuscript. Once again, you are presenting new ideas on a rather complex topic, would be a good idea to make the story flow..

Line 181, vaccinia virus, not vaccina virus

Author Response

Manuscript by Herbert et al summarizes data on Z-DNA, ZBP1, ADAR involvement, etc. Text is generally well-written and thoughtful. Based on background knowledge plus some new bioinformatic findings authors propose new model on ZBP-1 - HBV interactions which might have future therapeutic value. The idea is clearly interesting, is supported by the background and computational findings; whether it is really the case and will it work in a therapeutic setting, only more research will tell. Moreover, some of the authors views/terminology are somewhat controversial (e.g., what is meant by "digital genome" in the article's context?).

We expanded the description of the digital genome to address this question

Line 227 now reads”

“The sequences that flip to Z-DNA or Z-RNA under physiological conditions, named flipons, create binary switches where the DNA helix is either right- or left-handed, providing building blocks for a digital genome where responses are context specific and depend on the flipon conformation [67]. The flipons act by recruiting different sets of cellular machinery to DNA and RNA depending on its handedness, varying the readout of genetic information through their effects on transcript expression, editing and splicing.”

For the review/insight this is nevertheless OK (maybe manuscript would better fit into "Opinion" or similar article type though), however I do have a few concerns which has to be addressed.

1) This is submitted as a review, however some new data is presented. This is fine but has to be explained. I.e. what was done specifically and how. Otherwise there is no way to reproduce etc.

The analysis uses publicly available data and standard tools implicated in the R-package. We added these links to the legend of Figure 2

“The dataset for this analysis is available in singlecellexperiment format https://rdrr.io/github/LTLA/scRNAseq/man/ZhaoImmuneLiverData.html and can be analyzed using the protocols detailed at https://bioconductor.org/packages/release/bioc/vignettes/scater/inst/doc/overview.html”

We also added at the end of the manuscript

“Software: Data was analyzed as described in the legend for figure 2 using the R-project statistical language. We also added that code is available from the authors on reasonable request”

2) Lane 23 (and later in the manuscript) "Our focus is on Epstein-Barr virus (EBV), a24 pathogen that causes significant human morbidity and mortality." - This is an overstatement..

EBV does kill people either during the infectious mononucleosis stage or from cancer. The virus is also proposed an important cause of systemic lupus erythematosus and multiple sclerosis along with other autoimmune diseases. We have added the references from later in the paper at this earlier point in the discussion. The line now reads

“Our focus is on Epstein-Barr virus (EBV), a pathogen that causes significant human morbidity and mortality including autoimmune diseases like systemic lupus erythematosus, multiple sclerosis and cancer [18-20].”

Also for this, lane 208: "About 1-1.5% of worldwide cancer incidence are linked to the virus." - where does this statistics comes from?? 1 percent is a huge number.

Thanks for pointing this out -here is the reference that was omitted in the submission

Farrell, P.J. Epstein-Barr Virus and Cancer. Annu Rev Pathol 2019, 14, 29-53, doi:10.1146/annurev-pathmechdis-012418-013023.

3) Figures organization need improvements. For example, authors present their data on Fig. 2 and then jump back to Fig. 1 to present their model. This disrupts the flow of the paper. Fig. 4 has similar issues. Need to align text to figs throughout the manuscript. Once again, you are presenting new ideas on a rather complex topic, would be a good idea to make the story flow..

Thank-you for your helpful suggestion. You are completely correct. We have moved Figure 1C to create Figure 3, and Figure 4B to create Figure 7. All other figures have been renumbered.

We has also introduced a new title for the section discussing the new Figure 3

“A model for the effects of ZBP1 on T-cell Immunity” on line 99 to separate the data discussion from the interpretation.

And another one for the new figure 7

A model for EBV disease causation”  (line 479)

We also added to the abstract the following line to make it clear to the reader that this article was both a review and a preview, both based on existing data.

“The manuscript combines a review and reanalysis of existing data and provides an experimentally supported synthesis explaining roles for ZBP1 in T-Cell development and in chronic viral infections, using EBV as an example.”

Line 181, vaccinia virus, not vaccina virus

Corrected – thanks for catching this mistake.

Reviewer 2 Report

The manuscript deals with an important and timely subject. Many interesting results/issues are raised and reviewed within the manuscript, in a knowledgeable fashion . However, in its current form the manuscript is confusing as it includes data analysis and some stretches of speculation/hypothesis which are less fitting with a review format. For example, if the authors decide to make this a "data" paper, it would require restructuring and formal presentation of the "reclustering analysis" that was carried out, is presented in figures and serves as basis for throughts/considerations. Moreover, if the authors' intentions are to review + speculate (hypothesize), then I suggest clear separation between sections.

Author Response

The manuscript deals with an important and timely subject. Many interesting results/issues are raised and reviewed within the manuscript, in a knowledgeable fashion . However, in its current form the manuscript is confusing as it includes data analysis and some stretches of speculation/hypothesis which are less fitting with a review format. For example, if the authors decide to make this a "data" paper, it would require restructuring and formal presentation of the "reclustering analysis" that was carried out, is presented in figures and serves as basis for throughts/considerations. Moreover, if the authors' intentions are to review + speculate (hypothesize), then I suggest clear separation between sections.

Thanks for your positive comments. We did struggle with the questions you raised as the manuscript developed. While we started out with a review, there was sufficient data either missing or wrongly stated in the literature that we needed to supply this information. For example the transcription factors  listed in Figure 7 are derived from the single cell analysis performed rather than less precise literature data. We regard the paper as a synthesis and not a speculation as the presentation is data-driven and not conjecture due to the absence of any factual information. Also there are multiple hypothesis incorporated that can be subject to experimental verification beyond what we have validated using computational approaches.

We have gone with the last option you proposed where we  more clearly separate the data reviewed and analyzed from the models generated. We created two new model figures (as suggested by Reviewer 1) and two new sections outlining the synthesis of our findings. These sections are entitled”

“A model for the effects of ZBP1 on T-cell Immunity” (line 70)

“A model for disease when EBV defects”  (line 511)

We also added to the abstract the following insertion to make it clear to the reader that this article is more than a review.

“We describe and and provide a synthesis of the evidence.”

The synthesis leads to a new perspective with experimentally testable hypotheses.
